# The Potential Role of Gonadotropic Hormones and Their Receptors in Sex Differentiation of Nile Tilapia, *Oreochromis niloticus*

**DOI:** 10.3390/ijms26115376

**Published:** 2025-06-04

**Authors:** He Gao, Hongwei Yan, Tomomitsu Arai, Chak Aranyakanont, Shuang Li, Shigeho Ijiri

**Affiliations:** Division of Marine Life Sciences, Graduate School of Fisheries Sciences, Hokkaido University, Hakodate 041-8611, Hokkaido, Japan; gaohe920913@outlook.com (H.G.);

**Keywords:** Nile tilapia, FSH signaling, *cyp19a1a*, sex differentiation

## Abstract

Nile tilapia, as an ideal model for studying sex differentiation, is a popular farmed fish worldwide with a stable XX/XY sex-determination system. In tilapia, ovarian differentiation is triggered by estradiol-17β (E2) production in undifferentiated gonads. In a previous study, we suggested that follicle-stimulating hormone (FSH) signaling might be involved in ovarian differentiation in Nile tilapia. In this study, we further investigated the role of FSH signaling in ovarian differentiation via aromatase expression, which converts testosterone to E2. Masculinization of XX fry by aromatase inhibitor or 17α-methyltestosterone leads to suppression of *fshr* expression. Feminization of XY fry by E2 treatment increased *fshr* expression from 15 days after hatching, when E2 treatment was terminated. XX tilapia developed ovaries harboring aromatase expression if *fsh* and *fshr* were double knockdowns by morpholino-oligo injections. Finally, the transcriptional activity in the upstream region of the aromatase gene (*cyp19a1a*) was further increased by FSH stimulation when HEK293T cells were co-transfected with *foxl2* and *ad4bp*/*sf1*. Collectively, this study suggests that the role of FSH signaling is not critical in tilapia ovarian differentiation; however, FSH signaling may have a compensatory role in ovarian differentiation by increasing *cyp19a1a* transcription in cooperation with *foxl2* and *ad4bp*/*sf1* in Nile tilapia.

## 1. Introduction

Sex control in aquaculture can increase the production of target species relative to the limited amount of land, water, space, and food used, minimize environmental pollution, prevent the occurrence of diseases, improve management, and, finally, achieve the aquaculture industry’s sustainable development.

Nile tilapia (*Oreochromis niloticus*) is a gonochoristic teleost and has a stable XX/XY sex-determination system. In this species, phenotypic sex can be manipulated through hormonal treatment. All-female (XX) or all-male (XY) populations have been produced by crossing normal eggs (XX) with sex-reversed male sperm (XX) or super-male sperm (YY), respectively [1]. It is possible to obtain lots of monosex Nile tilapia, approximately 14 days, by this method. Therefore, Nile tilapia is an ideal model species for studying sex differentiation and sex determination.

The Genetically Improved Farmed Tilapia (GIFT) strain of Nile tilapia is popular in the world because of its higher growth performance [2]. Furthermore, gonadal maturation in Nile tilapia only takes 6–8 months, shorter than most farmed fish, and it is possible to spawn all year round in the laboratory [3]. The first morphological signs of gonadal sex differentiation appear in tilapia fry 23 to 26 days after hatching (dah), with the formation of the ovarian cavity in XX gonads or the efferent duct in XY gonads [4,5]. Molecular sex differentiation, however, occurs much earlier. In XX gonads, the transcription of ovary-type aromatase (*cyp19a1a*), an enzyme responsible for estradiol-17β (E2) synthesis, begins at 5 dah, coinciding with the upregulation of forkhead box L2 (*foxl2*), a transcription factor that activates *cyp19a1a* transcription by binding to its promoter region [1,6]. In undifferentiated XY gonads, gonadal soma-derived factor (*gsdf*) and doublesex and mab-3 related transcription factor 1 (*dmrt1*) genes exhibit male-superior expressions after 5 and 6 dah, respectively [1,7].

Moreover, endogenous E2 is a key steroid hormone, and aromatase is a key enzyme, in determining ovarian differentiation in tilapia, as observed in reptiles [8] and birds [9]. Estrogen administration can induce genetic males to develop as females [10]. The treatment of tilapia fry with letrozole (an aromatase inhibitor, AI) that blocks aromatase activity, results in complete sex change to functional males, an effect that can be reversed by simultaneous treatment with E2 [11,12]. However, the gene cascade that results in the female-specific expression of *cyp19a1a*, and subsequent ovarian differentiation, remains poorly understood in fish, including tilapia.

The tilapia *cyp19a1a* promoter region contains an *ad4bp*/*sf1* (nuclear receptor subfamily 5, group A, member 1, *nr5a1*) binding site, suggesting that *ad4bp*/*sf1* is a possible candidate for the regulation of *cyp19a1a* expression [13]. However, because *ad4bp*/*sf1* does not exhibit different expression levels between undifferentiated male and female gonads during early sex differentiation [1], its role in female-specific *cyp19a1a* expression appears limited. In contrast, *foxl2* exhibits a female-specific expression pattern [1] and may play a decisive role in ovarian differentiation by activating *cyp19a1a* transcription either directly or in conjunction with *ad4bp*/*sf1*, thereby initiating and increasing estrogen production [6].

In mammals, the primary factor that stimulates *cyp19a1a* expression and enzyme activity is follicle-stimulating hormone (FSH) [14,15,16,17]. In nonmammalian vertebrates, FSH receptor (FSHR) gene (*fshr*) expression is much higher in the ovary than in the testes and follows a pattern similar to that of *cyp19a1a* [18,19,20]. When FSH binds to FSHR, ATP is converted into cyclic adenosine monophosphate (cAMP), which in turn induces *cyp19a1a* mRNA expression in a dose-dependent manner in Japanese flounder (*Paralichthys olivaceus*), suggesting that FSH may play an important role in the upregulation of *cyp19a1a* during ovarian differentiation [19]. Moreover, in mammals, Sertoli cells in the testis and granulosa cells in the ovary express *fshr* [21], whereas, in fish, not only these cells but also Leydig cells in the testis and theca cells in vitellogenic ovarian follicle express *fshr* [22,23,24,25].

To investigate the potential role of gonadotropic hormones (GTHs) and their receptors (GTHRs) in sex differentiation in tilapia, the precise timing of the expression of FSH, luteinizing hormone (LH) in the pituitary gland, and their receptors (*fshr* and *lhr*) in the gonad were previously determined [20]. The expression patterns of *fsh* and *lh* in pituitaries, and *fshr* and *lhr* in gonads, were examined. In pituitaries, *fsh* expression began at 3 dah, with similar levels between XX and XY fish. In gonads, *fshr* expression began at 5 dah and was significantly higher in XX than XY between 6 and 25 dah. In contrast, *lh* expression in pituitaries began at 25 dah without sexual dimorphism. Although the expression levels of *lhr* from 10 to 25 dah were higher in XX gonads than XY gonads, the sexual dimorphism pattern of *lhr* appeared later than that of *fshr*. These results suggest that LH signaling has a limited role in gonadal differentiation, while FSH signaling may be involved in ovarian differentiation in Nile tilapia [20]. Moreover, in fish, it is well accepted that LH is responsible for oocyte maturation [26,27,28].

Little is known about the relationship between FSH signaling and *cyp19a1a* transcription during ovarian differentiation in tilapia. It is critical to find out more information about the mechanism of sex differentiation and determination to provide theoretical and technical methods to contribute to environmentally friendly, resource-saving, quality-safe, and sustainable aquaculture systems. Therefore, in the present study, we investigated whether high *fshr* expression levels in XX gonads contribute to the female-dominant expression of *cyp19a1a*, which initiates ovarian differentiation. Previous findings have demonstrated that treating genetic females with a non-steroidal aromatase inhibitor (letrozole) or an androgen (17α-methyltestosterone, MT) before and during gonadal sex differentiation causes female masculinization (sex reversal from female to male). Conversely, estrogen treatment of genetic males induces phenotypic gonadal sex reversal to ovaries [10,29,30]. Female-to-male and male-to-female transdifferentiations are useful experimental models for investigating gonadal differentiation in Nile tilapia.

Therefore, we first examined the effects of masculinization and feminization processes, particularly the absence or presence of E2, on changes in GTHR expression profiles in the gonads during sex reversal induced by AI, MT, and E2. Furthermore, to examine whether disruption of FSH signaling influences *cyp19a1a* expression, E2 production, and ovarian differentiation, morpholino (MO)-mediated knockdown of *fshr* and *fsh* gene expression was conducted. Finally, to investigate the functions of FSH signaling in sex differentiation in Nile tilapia, the relationship among FSH signaling, *foxl2*, *ad4bp*/*sf1*, and *cyp19a1a* was explored through a *cyp19a1a* promoter assay.

## 2. Results

### 2.1. Effects of AI and MT on the Expression Profiles of fshr and lhr mRNA in XX Fry Gonads

Similarly to the results of a previous study [20], *fshr* mRNA was initially detected at similar levels in control XX and XY gonads at 5 dah (*p* > 0.05), but its expression in control XX gonads was significantly higher than in control XY gonads from 6 to 25 dah (*p* < 0.05). Both AI and MT treatments resulted in a significant decrease in *fshr* mRNA expression between 12 and 25 dah (*p* < 0.05). At 40 dah, *fshr* mRNA levels were similar among control XX, XY, AI-treated, and MT-treated fry gonads (*p* > 0.05) (Figure 1a).

After MT treatment, *lhr* mRNA expression was suppressed in comparison with that in the control XX gonads from 12 dah onward and reached levels similar to those in the control XY gonads after 20 dah. Interestingly, AI treatment also downregulated the expression of *lhr* mRNA, but a delayed trend was observed compared with the MT treatment. *lhr* mRNA expression in AI-treated XX tilapia gonads remained similar to that in the control XX group before 20 dah; thereafter, suppression occurred after 20 dah. At 40 dah, the levels of *lhr* mRNA were similar in control XY and MT- and AI-treated gonads, and significantly higher than those in control XX gonads (Figure 1b).

### 2.2. Effects of E2 on the Expression Profiles of fshr and lhr mRNA in XY Fry Gonads

After E2 treatment from 4 dah, the *fshr* mRNA expression level did not immediately increase in the E2-treated XY fry group, but exhibited a roughly similar expression pattern to that in the control XY male group from 5 to 15 dah (*p* > 0.05), which was significantly lower than that in the control XX female group during this period (6–15 dah). After the E2 treatment finished, *fshr* mRNA expression rapidly increased from 25 dah in the E2-treated XY fry group, reaching similar levels to those in the control XX female group at 25 and 40 dah. At 25 dah, significantly higher *fshr* mRNA expression levels were detected in both the control XX female and E2-treated XY fry groups compared to the control XY male group (*p* < 0.05). *fshr* mRNA expression was similar across all three groups at 40 dah (*p* > 0.05) (Figure 2a).

Following feminization with E2, similar changes were observed in *lhr* mRNA expression patterns. During 5 to 15 dah, *lhr* mRNA expression levels in E2-treated XY fry were not elevated in comparison to those in the control XX fry group. *lhr* mRNA levels in E2-treated XY fry increased rapidly from 25 dah, reaching levels similar to those in control XX fry at 25 and 40 dah (*p* > 0.05) (Figure 2b).

### 2.3. Specific Depletion of a Targeted Protein in Nile Tilapia Using MO

As shown previously [20], expression of the FSH protein can be detected in the pituitary gland as early as 3 dah. To check the efficacy of the microinjection method, FSH protein was immunohistochemically analyzed after injection with *fsh*-MO. A few FSH-immunoreactive cells were detected in control XX female pituitary glands; in contrast, no FSH protein was detected in *fsh*-MO-injected abnormal fry or most normal fry pituitary glands (80%) at 4 and 7 dah (Figure 3(B2,D2)). At 10 dah, FSH protein was detected again in the pituitary glands of *fsh*-MO-injected fry. Because of low *fshr* expression during early sex differentiation and the unavailability of an anti-tilapia FSHR antibody, the depletion of FSHR by knockdown could not be verified.

### 2.4. Effect of MO Injection on the Expression of Aromatase and gsdf in XX Gonads and Ovarian Differentiation

At 10 and 25 dah, aromatase-immunoreactive cells were observed in both the female XX control (DW control and MO control) gonads (Figure 4 and Figure 5) and the *fshr*-MO, *fsh*-MO, and *fsh* + *fshr*-MO-injected XX gonads (Figure 4 and Figure 5), whereas no immunoreactive cells were observed in the control XY male gonads. GSDF protein was detected in the male XY control gonads, while no GSDF-immunoreactive cells were observed in the XX female control, *fshr*-MO, *fsh*-MO, or *fsh* + *fshr*-MO injected XX gonads (Figure 4 and Figure 5). The same results were obtained at 7 dah.

At 40 or 50 dah, ovarian differentiation was assessed using hematoxylin and eosin staining. The efferent duct was observed in control XY male gonads, whereas the ovarian cavity and oocytes were seen in control female gonads and MO-injected female gonads. Compared with the control groups, sex reversal was not observed in any individuals in the MO groups, and normal ovarian differentiation occurred.

### 2.5. FSH Signaling Alone Did Not Activate cyp19a1a Expression in HEK293T

When *fshr* alone was transfected into HEK293T cells followed by incubation with recombinant FSH protein (rFsh), increasing concentrations of rFsh did not induce relative luciferase activity (Figure 6). At high concentrations of rFsh (200 and 400 ng/mL), relative luciferase activities were significantly lower compared to those at low concentrations (50 and 100 ng/mL) (Figure 6).

### 2.6. FSH Signaling Interacts with foxl2 and ad4bp/sf1 to Activate cyp19a1a Expression in HEK293T

In HEK293T cells, *ad4bp*/*sf1* alone could activate *cyp19a1a* expression (Figure 7, group 4), whereas rFsh with *fshr* alone or *foxl2* alone did not induce *cyp19a1a* expression compared to the control (Figure 7, groups 1, 2, and 3). Moreover, rFsh with *fshr* and *foxl2* also did not activate *cyp19a1a* expression (Figure 7, group 9).

*foxl2* enhanced *ad4bp*/*sf1*-activated *cyp19a1a* expression when HEK293T cells were co-transfected with *ad4bp*/*sf1* (Figure 7, group 5). rFsh with *foxl2* or *fshr* did not enhance *ad4bp*/*sf1*-activated *cyp19a1a* expression when HEK293T cells were co-transfected with *ad4bp*/*sf1* (Figure 7, groups 7 and 8). However, rFsh with *fshr* enhanced the *foxl2*- and *ad4bp*/*sf1*-activated *cyp19a1a* expression when HEK293T cells were co-transfected with *foxl2* and *ad4bp*/*sf1* (Figure 7, groups 5, 10, 11, 12, and 13).

Although there was no significant difference between HEK293T cells co-transfected with *foxl2*, *ad4bp*/*sf1*, and *fshr* and incubated with 50 ng/mL rFsh compared to those without rFsh (Figure 7, groups 6 and 10), increasing the concentration of rFsh (to 200 and 400 ng/mL) stimulated *cyp19a1a* expression (Figure 7, groups 6, 12 and 13), although the differences were not statistically significant. Therefore, *cyp19a1a* expression was stimulated most strongly when *foxl2*, *ad4bp*/*sf1*, and *fshr* were co-transfected into HEK293T cells with rFsh incubation.

## 3. Discussion

The expressions of both *fshr* mRNA and *lhr* mRNA were suppressed by both AI and MT treatment in XX fry. However, they exhibited different expression patterns during masculinization. *fshr* mRNA expression levels were downregulated immediately after MT and AI treatment. The rapid suppression of *fshr* expression in MT-treated XX gonads suggests that exogenous androgen might directly downregulate *fshr* expression or indirectly suppress it in the absence of E2 synthesis. The hypothesis that MT may indirectly downregulate *fshr* expression is supported by the fact that MT induces masculinization and suppresses the expression of key steroidogenic enzymes, including aromatase throughout sex differentiation in tilapia [31].

After AI treatment, *fshr* expression was also rapidly suppressed, indicating that the absence of E2 may cause the downregulation of *fshr* expression. Taken together, these results suggest that suppression of E2 synthesis may induce the downregulation of *fshr* mRNA expression during the masculinization process. Therefore, *fshr* expression may be dependent on the E2 signal. Moreover, the possibility that the downregulation of *fshr* may have contributed, at least in part, to the loss of estrogen synthesis after MT or AI treatment cannot be excluded. Changes in the expression profiles of *fshr* mRNA during the masculinization process also suggest that there might be positive feedback between FSH signaling and E2, and that *fshr* could be involved in the positive regulation of *cyp19a1a* expression and E2 production in Nile tilapia.

Interestingly, *fsh* is also downregulated during masculinization in rainbow trout (*Oncorhynchus mykiss*) [32], demonstrating that *fsh* is another important candidate gene as an early regulator of ovarian estrogen production. Because no sexual dimorphism in *fsh* mRNA expression has been observed during the early sex differentiation period in Nile tilapia [20], changes in *fsh* mRNA expression were not investigated in this study. In contrast to rainbow trout, *fshr*, rather than *fsh*, might be involved in sex differentiation in Nile tilapia. Nevertheless, FSH protein was detected in the pituitary gland as early as 3 dah during sex differentiation; therefore, it is possible that *fsh* indirectly affects sex differentiation in Nile tilapia.

Following feminization by E2 treatment, high *fshr* mRNA expression levels were not observed in E2-treated XY gonads until 15 dah, compared with those in control XX gonads. These results appear to contradict the initial assumption (based on the MT and AI masculinization experiments) that a decrease in E2 production would rapidly suppress *fshr* expression levels. This could have been caused by the rather unusual physiological condition of high E2 concentration. Therefore, exogenous estrogens may not be able to induce *fshr* expression before 15 dah.

In trout, it has been found that even if estrogens can induce a strong and rapid upregulation of *foxl2*, they are unable to restore *cyp19a1* expression [32]. Moreover, at the end of treatment when the gonads were feminized, *cyp19a1* was slowly upregulated, supporting the idea that the restoration of *cyp19a1* expression is blocked by exogenous estrogen treatment [32]. Therefore, in the present study, it is possible that the strong effect of exogenous estrogen may have blocked the induction of *cyp19a1a* expression and endogenous estrogen had not been synthesized, or not to a sufficient extent, to induce high *fshr* expression before E2 treatment terminated (4–10 dah). After 15 dah, endogenous estrogen synthesis may have provided sufficient E2 for the upregulation of *fshr* expression; consequently, a female *fshr* expression pattern was observed.

After MT treatment, *lhr* mRNA expression at 10–25 dah was rapidly suppressed. However, unlike *fshr*, the suppression of *lhr* expression was not observed until 20 dah in AI-treated gonads and occurred in a delayed pattern compared with that in MT-treated gonads. This suggests that exogenous androgens may directly downregulate *lhr* expression, but *lhr* expression may not be affected by E2 production inhibition, as indicated by the delayed suppression pattern observed after AI treatment. Moreover, it is possible that changes in *lhr* mRNA expression in AI-treated gonads after 25 dah were a result of the feminization of the whole steroid hormone system in sex-reversed fry.

Following feminization by E2 treatment, similar results were observed as with *fsh* mRNA. In our previous study, *lh* mRNA was not detected in the pituitary glands of either sex until 25 dah. Taken together, these results suggest that LH and LHR do not play a role in gonadal sex differentiation in Nile tilapia. FSH signaling might be involved in the regulation of *cyp19a1a* expression and E2 synthesis, and there may be a feedback relationship between FSH signaling and E2.

Expression of the FSH protein was not detected in most of the larval pituitary glands at 7 dah, indicating that the MO microinjection successfully reduced levels of the target FSH protein. The data also indicate that gene knockdown continued to at least 7 dah. Unlike *fsh*, it was impossible to check the effectiveness of the *fshr*-MO microinjection, because the *fshr* mRNA expression level in the gonads was low and an anti-tilapia FSHR antibody was unavailable. Similarly to control XX female tilapia, the XX individuals injected with *fshr*-MO exhibited normal aromatase expression without *gsdf* expression in the gonads at 7, 10, and 25 dah. At 40 or 50 dah, morphological observations revealed that normal ovarian differentiation occurred in the gonads of XX female tilapia injected with *fshr*-MO. *fshr* knockdown did not suppress aromatase expression, resulting in normal E2 production and ovarian differentiation. Moreover, no *gsdf* expression was detected in the gonads of *fshr*-MO-injected XX tilapia, demonstrating that the presence of E2 inhibits male-specific *gsdf* expression. It is possible that a single knockdown of *fshr* might not be sufficient to completely disrupt FSH signaling. Therefore, we conducted a knockdown of *fsh* and a double knockdown of *fshr* + *fsh* in the present study. However, this also did not stimulate any changes in the expression of aromatase or *gsdf* in the gonads, and sex-reversed larvae were not observed. In addition, because FSH protein could still be detected in *fsh*-MO-injected larvae at 10 dah, it is possible that incomplete knockdown of FSH signaling after 10 dah may have resulted in normal ovarian differentiation.

In the previous study, *foxl2* alone had no effect on the transcription of *cyp19a1a* in HEK293T cells, while *foxl2* alone could induce the transcription of *cyp19a1a* in the mouse testicular line TM3. This difference might be because TM3 is a steroid-producing cell with endogenous *ad4bp*/*sf1* [6]. Although a longer *cyp19a1a* fragment was subcloned into the plasmid and genes were subcloned into the pSI vector, but not the pcDNA3.1 vector, similar results were observed.

In this study, FSH signaling, with or without *foxl2*, did not affect the *cyp19a1a* promoter activity in HEK293T cells. However, it has been reported that FSH signaling and *foxl2* are involved in the transcriptional regulation of *cyp19a1a* in Japanese flounder [19]. Therefore, the different results might be due to the deficiency of *ad4bp*/*sf1*. In addition, in rat ovaries, both cAMP and *ad4bp*/*sf1* regulate the expression of *cyp19a1a* [33,34]. In this study, the presence of FSH signaling and *ad4bp*/*sf1* could stimulate the *cyp19a1a* promoter activity. Considering the presence of several cAMP-responsive elements in the *cyp19a1a* promoter region in Nile tilapia [13], it suggests that FSH signaling and *ad4bp*/*sf1* play roles in *cyp19a1a* transcription in Nile tilapia. Moreover, rFsh with *fshr* could stimulate *foxl2*- and *ad4bp*/*sf1*-activated *cyp19a1a* expression when HEK293T cells were co-transfected with *foxl2* and *ad4bp*/*sf1*. Therefore, FSH signaling can stimulate *cyp19a1a* expression with *ad4bp*/*sf1* and *foxl2*. Considering the expression patterns of sex-differentiation-related genes, such as *cyp19a1a*, *foxl2*, and *ad4bp*/*sf1* [1], in the undifferentiated gonads of genetic female Nile tilapia during molecular sex differentiation, the expression levels of *foxl2* and *fshr* are higher than those in genetic male Nile tilapia, and the transcription activity of *cyp19a1a* is induced in the presence of *ad4bp*/*sf1*, higher *foxl2*, and higher FSH signaling.

In genetic male Nile tilapia, although *ad4bp*/*sf1* is present, the transcription activity of *cyp19a1a* fails to be upregulated because of the deficiency of *foxl2*, lower FSH signaling, and the presence of male-specific genes such as the Y-localized anti-Müllerian hormone (*amhy*), *dmrt1*, and *gsdf*, which downregulate the transcription activity of *cyp19a1a* [35,36]. After rFsh injection, the *cyp19a1a* expression level in the undifferentiated gonads of XX individuals tended to be stimulated [37]. Therefore, FSH signaling may interact with *foxl2* and *ad4bp*/*sf1* to promote *cyp19a1a* transcription.

In mammals, knocking out *fshb* or *lhb* does not affect early gonadal development, though it does reduce the number of germ cells [38,39]. However, a complete lack of FSHR signaling results in estrogen deprivation in female mice [40]. In two teleost fishes, the medaka [41] and zebrafish [42], loss of FSHR function leads to masculinization. Therefore, as knockout technology can completely eliminate FSH signaling, it is likely that FSH signaling-deficient tilapia will be generated in the near future to confirm the results obtained previously. Further research is needed to clearly characterize the role of FSH signaling in ovarian differentiation.

In the previous study, the expression pattern of *fshr* during molecular sex differentiation suggests that FSH signaling may be involved in ovarian differentiation [20]. Although XX tilapia developed ovaries harboring aromatase expression if *fsh* and *fshr* were double knockdown by morpholino-oligo injections, the transcriptional activity in the upstream region of *cyp19a1a* was increased by FSH stimulation when HEK293T cells were co-transfected with *foxl2* and *ad4bp*/*sf1*. In summary, the results of this study indicate that FSH signaling may play a compensatory role in ovarian differentiation and ovarian development by regulating *cyp19a1a* transcription by interacting with *foxl2* and *ad4bp*/*sf1* in Nile tilapia.

## 4. Materials and Methods

### 4.1. Experimental Fish

Adult Nile tilapia (sex-reversed males XX, normal females, normal males, and super-males YY) were kept in freshwater (26 °C) under a 14L:10D photoperiod in indoor re-circulating freshwater tanks and were fed commercial trout pellets ad libitum (Marubeni Nisshin Feed Co., Ltd., Tokyo, Japan). Under these conditions, females spawned repeatedly every 14 to 18 days. Eggs were stripped on the day of spawning and fertilized by the usual drying method. All-female (XX) or all-male (XY) populations were obtained by crossing a normal female (XX) with a sex-reversed male (XX) or a super-male (YY), respectively, as described elsewhere [1]. Fertilized eggs cultured in 50 mL round-bottom glass tubes were kept rotating by injecting circulating water at 26 °C. Fry hatched after 4 days. The fry were randomly distributed into 60 L glass tank containers for the control and treatment groups. At 5 dah, the fry were transferred to a 60 L aquarium. The fry were fed with commercial food (with or without treatments) starting at 9 dah (Marubeni Nisshin Feed Co., Ltd.).

### 4.2. Treatment of XX Fry with AI or MT

Treatment with a non-steroidal AI (letrozole; LKT Laboratories Inc., St. Paul, MN, USA) or an androgen (MT; Sigma-Aldrich, St. Louis, MO, USA) was conducted from the first feeding day (9 dah) to 20 dah in the all-female XX populations. AI or MT was administered by adding it to the food using the ethanol evaporation method (AI, 200 mg/kg food; MT, 10 mg/kg food), and AI- or MT-treated food was fed to the larvae. Briefly, AI or MT was dissolved in 100% ethanol and the solution was sprayed over the food. A 100% masculinization of XX larvae was induced by treating them 4 times per day from 9 to 20 dah. After the treatments, the normal diet was resumed.

### 4.3. Immersive Treatment of XY Fry with E2

To obtain XY sex reversal fry, XY fry were immersed in water containing E2 (Sigma-Aldrich) at a final concentration of 900 ng/mL from 4 to 10 dah. The water was renewed daily, and 100% feminization was successfully induced.

### 4.4. Quantitative Reverse Transcription-Polymerase Chain Reaction (qRT-PCR)

Gonads from the XX female control group and XY male control group were collected at 5, 6, 7, 10, 12, 15, 20, 25, and 40 dah; those from the AI-treated XX fry group and the MT-treated XX fry group were collected at 10, 12, 15, 20, 25, and 40 dah; and those from the E2-treated XY fry group were collected at 5, 6, 7, 10, 15, 25, and 40 dah. The number of fish sampled from each group was as follows: for each sampling point between 5 and 12 dah, 60 to 80 larvae; 15 to 25 dah, 30 to 50 fry; and at 40 dah, at least 20 fry. The fish were dissected under a stereoscopic microscope (Leica, Singapore), and the viscera were removed. RNA later reagent (Ambion, Austin, TX, USA) was added to the coelomic epithelium to stabilize the RNA in the gonads. The gonads were then removed using fine forceps. Gonads collected at each sampling point were pooled in a tube with 200 µL of RA1 buffer and 4 µL of tris (2-carboxyethyl) phosphine (NucleoSpin RNA XS; Macherey-Nagel, Duren, Germany), and were immediately stored at −80 °C until RNA extraction. Sampling was independently repeated at least three times at each sampling point. The primer sets are listed in Table 1.

Total RNA was extracted from the gonads using an RNA extraction kit (NucleoSpin RNA XS) and quantified with a NanoDrop spectrometer (NanoDrop Technologies Inc., Wilmington, DE, USA). The quantified RNA samples were subjected to a one-step reverse transcription polymerase chain reaction (RT-PCR) using an ABI Prism 7300 unit (Applied Biosystems, Foster City, CA, USA). Reactions were performed with 10 ng of total RNA in a final volume of 20 L that contained 10.4 µL 2X SYBR Green Reaction Mix with ROX at a final concentration of 500 nM, 0.4 µL SuperScript III RT/Platinum Taq mix, and 4 μM of each primer using a one-step RT-PCR kit (SuperScript III Platinum SYBR Green One-Step qRT-PCR Kit; Invitrogen, Carlsbad, CA, USA). The cycling program was as follows: 50 °C for 3 min hold, 95 °C for 5 min hold, and 40 cycles of amplification (95 °C for 15 s, 60 °C for 30 s). A melting-curve analysis was performed for each reaction to confirm single amplification. Data from the qRT-PCR are expressed as the mean ± standard error of the mean (SEM) of at least three independent samples.

### 4.5. MO Injections

The MOs were designed to be targeted at the transcriptional initiation site of *fsh* (GenBank accession No. AY294015) or *fshr* (GenBank accession No. AB041762) to interfere with Nile tilapia *fsh* or *fshr* transcription, respectively. For MO control, a standard control oligo was used. The MOs, along with the MO control (300 nmol), were purchased from Gene Tools (Philomath, OR, USA), and their sequences are shown in Table 1. A working solution was prepared by diluting the MO solution in sterile water with a final concentration of 0.025 to 0.050% phenol red (as an injection indicator). Firstly, approximately 25 nL of *fshr*-MO was injected into 1–2-cell-stage genetic XX embryos. A *fshr*-MO control was injected into XX embryos, and distilled water (DW) was injected into embryos from the all-XX and all-XY populations as negative controls. Subsequently, approximately 25 nL of *fsh*-MO or co-injection (*fsh*-MO and *fshr*-MO) was injected into 1–2-cell-stage genetic XX embryos. Distilled water (DW) was injected into embryos from the all-XX and all-XY populations as negative controls. After microinjection, all embryos were placed onto microplates (Asahi Techno Glass Corp., Chiba, Japan) with freshwater and cultured on a shaker at 26 °C. Over the next hours and days of development, the embryonic and larval phenotypes were observed under a microscope. Fry hatched 4 days after fertilization (daf). At 5 dah, the fry were transferred to a 60 L aquarium tank. The fry were fed with commercial food (with or without treatments) starting at 9 dah (Marubeni Nisshin Feed Co., Ltd.).

### 4.6. Tissue Collection and Immunohistochemistry

To confirm the success of the microinjection, the heads of larvae were collected from the control XX female group (injected with DW) and the XX female group (injected with *fsh*-MO, 0.87µM/embryo) at 3, 4, 5, 7, and 10 dah. To investigate the effects of a single knockdown of *fshr* or *fsh*, and a double knockdown of both *fshr* and *fsh* on ovarian differentiation, gonads attached to the trunk were dissected at 7, 10, 25, and 40 dah (or 50 dah). The tissues were then subjected to immunohistochemical analysis and hematoxylin–eosin staining.

For the immunohistochemical analysis, the heads or gonads were fixed in Bouin’s solution for 24 h, dehydrated in an ascending series of ethanol, and embedded in paraffin. Serial sections of 6 µm thickness were mounted on silanized slides (Matsunami Glass Ind. Ltd., Osaka, Japan) and subjected to immunohistochemical staining using a Histofine kit (Nichirei Biosciences Inc., Tokyo, Japan). To examine FSH protein in the pituitary, aromatase, and GSDF in the gonads, anti-Nile tilapia FSH antibody [18], anti-Nile tilapia aromatase antibody [8], and anti-Nile tilapia GSDF antibody [5] were used as the primary antibodies, respectively, according to the Histofine SAB-PO (R) kit manufacturer’s protocol (Nichirei Bioscience, Tokyo, Japan).

### 4.7. Construction of Nile Tilapia cyp19a1a Promoter Reporter Plasmid and Expression Plasmids for Nile Tilapia foxl2 and ad4bp/sf1

A Nile tilapia *cyp19a1a* gene promoter fragment of approximately −3066 bp was generated by PCR from Nile tilapia genomic DNA. The fragment was PCR-amplified using KOD FX Neo (Toyobo, Osaka, Japan) according to the manufacturer’s instructions. The PCR for the *cyp19a1a* gene promoter fragments was performed in 20 μL reaction mixtures containing 0.4 μL of KOD FX Neo (1 U/μL), 10 μL of 2 × PCR buffer for KOD FX Neo, 4 μL of 2 mM dNTP, 0.6 μL of 10 mM forward primer, 0.6 μL of 10 mM reverse primer, 1 μL of template genomic DNA of 25 dah ovary, and 3.4 μL of distilled water. The PCR conditions were as follows: an initial denaturation step of 94 °C for 2 min, followed by 35 cycles of 98 °C for 10 s, 55 °C for 30 s, and 68 °C for 95 s. The 5′ flanking region of the Nile tilapia *cyp19a1a* was inserted into the pGL3-basic vector (Promega Corp., Madison, WI, USA) at the SmaI site using Gibson assembly to create the *cyp19a1a*-luciferase reporter vector. The primer set is listed in Table 1.

Ovaries from genetic XX female Nile tilapia were sampled, and total RNA was extracted using the NucleoSpin RNA XS kit (Macherey-Nagel, Duren, Germany). Subsequently, cDNA was synthesized from 0.5 µg total RNA using Revertra Ace reverse transcriptase (Toyobo Co., Ltd., Osaka, Japan). The reaction was conducted in a 20 µL volume containing 1.5 µL of 10 mM Oligo d(T) Primer, 2 µL of dNTP mix (10 mM each), 4.0 µL of 5 × RT buffer, and 1.0 µL of Revertra Ace. The process included an initial incubation at 30 °C for 10 min, followed by an extension at 42 °C for 90 min. Afterward, the mixture was incubated at 99 °C for 5 min to inactivate the reverse transcriptase, and then 40 µL of TE (pH 8.0) was added.

The full-length open reading frames encoding Nile tilapia *foxl2* and *ad4bp*/*sf1* were amplified by PCR with gene-specific primers (listed in Table 1) from the cDNA of XX Nile tilapia ovaries. Each gene was PCR-amplified using KOD-Plus-Neo (Toyobo) according to the manufacturer’s instructions. Briefly, a 50 μL reaction mixture was prepared with the following reagents: 5 μL of 10 × PCR Buffer for KOD–Plus-Neo, 5 μL of 2 mM dNTPs, 3 μL of 25 mM MgSO_4_, 1.5 μL of 10 μM each primer, 1 μL of template cDNA from 25 dah ovary, 1 μL of KOD-Plus-Neo (1 U/μL), and 32 μL of distilled water. The PCR conditions were as follows: pre-denaturation at 94 °C for 2 min, followed by 35 cycles of amplification consisting of denaturation at 98 °C for 10 s, annealing at 57 °C for 30 s, and extension at 68 °C for 30 s (*foxl2*), or 40 s (*ad4bp*/*sf1*). The transcription factors *foxl2* and *ad4bp*/*sf1* were subcloned into the pSI plasmid ECoRI site by Gibson assembly to generate pSI-*foxl2* and pSI-*ad4bp*/*sf1* expression vectors. The primer sets are listed in Table 1. The pSI-*fshr* plasmid was produced in a previous study [27].

### 4.8. Cell Culture, Transfection, and cyp19a1a Promoter Assay by Dual-Luciferase Assay

HEK293T cells were seeded at a density of 1 × 10^5^ cells into 24-well culture plates (Corning Inc., Corning, NY, USA) and grown in DMEM (FBS+) with 5% CO_2_ at 37 °C. The next day, the cells were transfected using ScreenFactTM A plus (Fujifilm Wako Pure Chemical Corporation, Richmond, VA, USA) with the following plasmids: 0–100 ng of pSI plasmid containing cDNAs encoding *foxl2*, *ad4bp*/*sf1*, and *fshr*; 500 ng of pGL3 plasmid with *cyp19a1a* promotor; and 100 ng of pRL-TK plasmid for relative quantification. After transfection, the HEK293T cells were incubated with different concentrations of rFsh (0, 50, 100, 200, and 400 ng/mL), which was produced in a previous study [27], for 48 h. After incubation, firefly and *Renilla* luciferase activities were measured using the Dual-Luciferase Reporter Assay System (Promega, Madison, WI, USA) and a Luminescencer PSN AB-2200 (ATTO Co., Ltd., Tokyo, Japan). *Renilla* luciferase activity served as an internal control, and firefly luciferase activity was normalized to *Renilla* luciferase activity.

### 4.9. Statistical Analyses

Results are reported as the mean ± standard error (SEM). Significant differences were determined using Tukey–Kramer tests and Student’s *t*-test, performed with Excel Statistical Analysis Ver. 7.0 (ESUMI Co., Ltd., Tokyo, Japan). A statistically significant difference was indicated by a *p* value of <0.05.

## Figures and Tables

**Figure 1 ijms-26-05376-f001:**
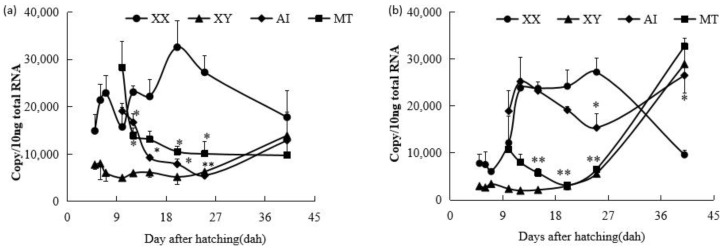
Expression profiles of *fshr* (**a**) and *lhr* mRNA (**b**) in control XX female fry, control XY male fry, letrozole (aromatase inhibitor, AI)-treated XX fry, and 17α-methyltestosterone (MT)-treated XX fry gonads during gonadal development from 5 to 40 days after hatching (dah). Each value represents the mean ± standard error of the mean (SEM) of three measurements. Circles represent means for XX groups, triangles represent means for XY groups, diamonds represent means for AI-treated XX fry groups, and squares represent means for MT-treated XX fry groups. **, Significant difference between the control XX and hormone-treated XX groups (*p* < 0.01); *, significant difference between the control XX and hormone-treated XX groups (*p* < 0.05) by Student’s *t*-test.

**Figure 2 ijms-26-05376-f002:**
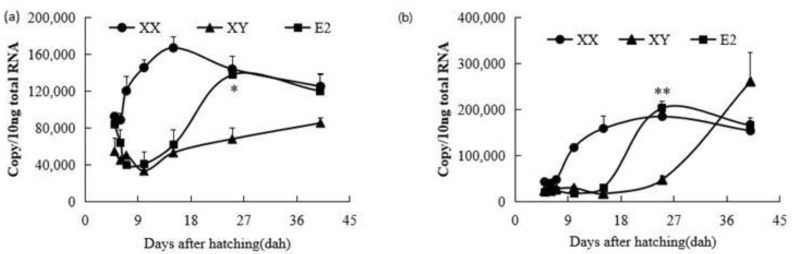
Expression profiles of *fshr* (**a**) and *lhr* mRNA (**b**) in control XX female fry, control XY male fry, and estradiol-17β (E2)-treated XY fry gonads during gonadal development from 5 to 40 days after hatching (dah). Each value represents the mean ± standard error of the mean (SEM) from three measurements. Circles represent means for XX groups, triangles represent means for XY groups, and squares represent means for E2-treated XY fry groups. **, Significant differences between the control XY and E2-treated XY groups *(p* < 0.01); *, significant differences between the control XY and E2-treated XY groups (*p* < 0.05) by Student’s *t*-test.

**Figure 3 ijms-26-05376-f003:**
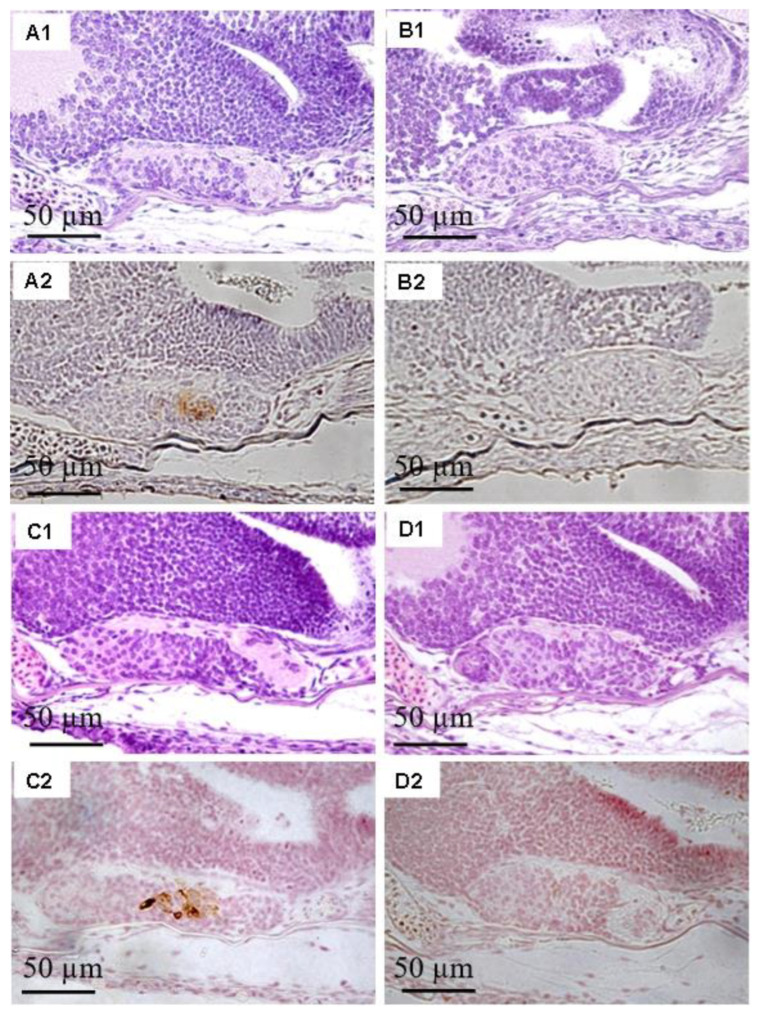
FSH protein in the pituitary glands at 4 and 7 days after hatching (dah). FSH protein was detected in the pituitary glands of XX control female fry injected with distilled water ((**A**), 4 dah; (**C**), 7 dah). No FSH protein was observed in the pituitary glands of MO-injected abnormal fry or most of the MO-injected normal fry ((**B**), 4 dah; (**D**), 7 dah). (**1**) hematoxylin–eosin staining; (**2**) immunohistochemical staining. Bars = 50 µm.

**Figure 4 ijms-26-05376-f004:**
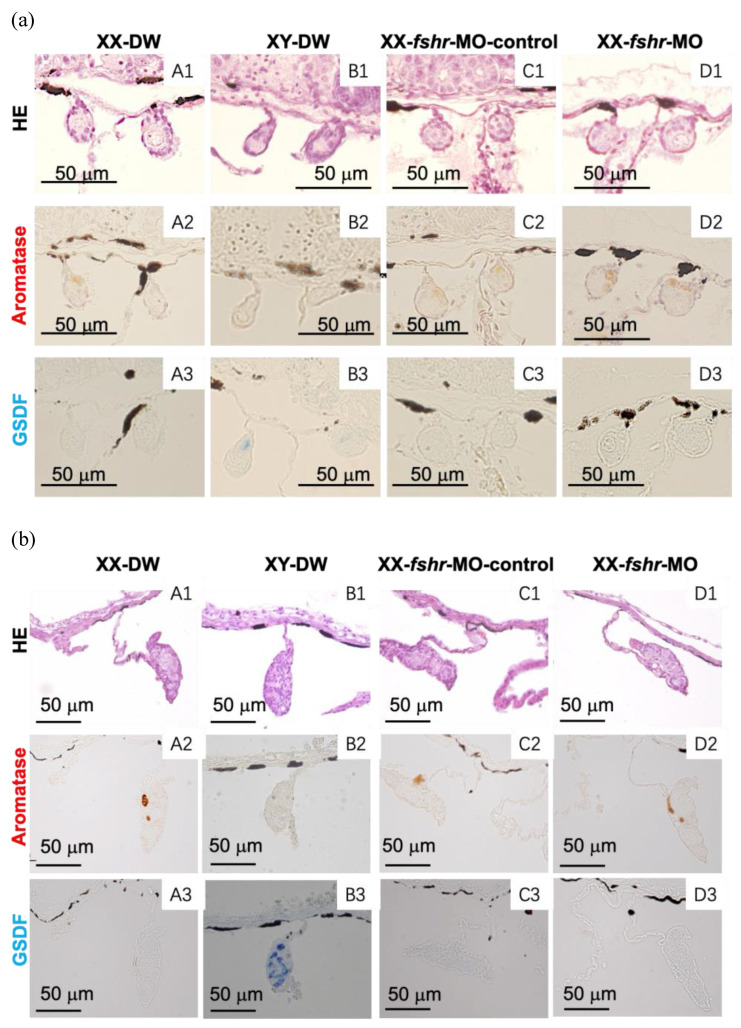
Effect of a single injection of distilled water (DW) (**A**,**B**), *fshr*-morpholino control (**C**), or *fshr* morpholino (**D**) on the expression of aromatase and GSDF in the gonads at 10 days after hatching (dah) (**a**) and 25 dah (**b**); HE, hematoxylin–eosin staining. (**1**) HE; (**2**,**3**) immunostaining of the gonads with anti-aromatase (**2**) or GSDF (**3**) serum. Bars = 50 µm.

**Figure 5 ijms-26-05376-f005:**
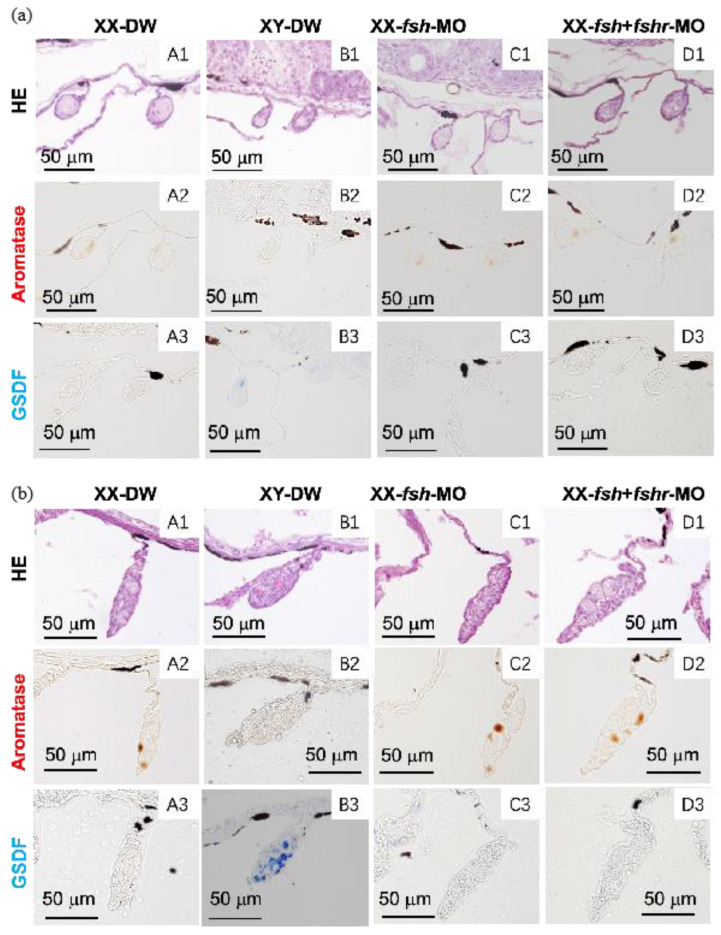
Effects of an injection of distilled water (DW) (**A**,**B**), *fsh* morpholino (**C**), or a co-injection of *fsh* and *fshr* morpholino (**D**) on the expression of aromatase and GSDF in the gonads at 10 days after hatching (dah) (**a**) and 25 dah (**b**); HE, hematoxylin–eosin staining. (**1**) HE; (**2**,**3**) immunostaining of the gonads with anti-aromatase (**2**) or GSDF (**3**) serum. Bars = 50 µm.

**Figure 6 ijms-26-05376-f006:**
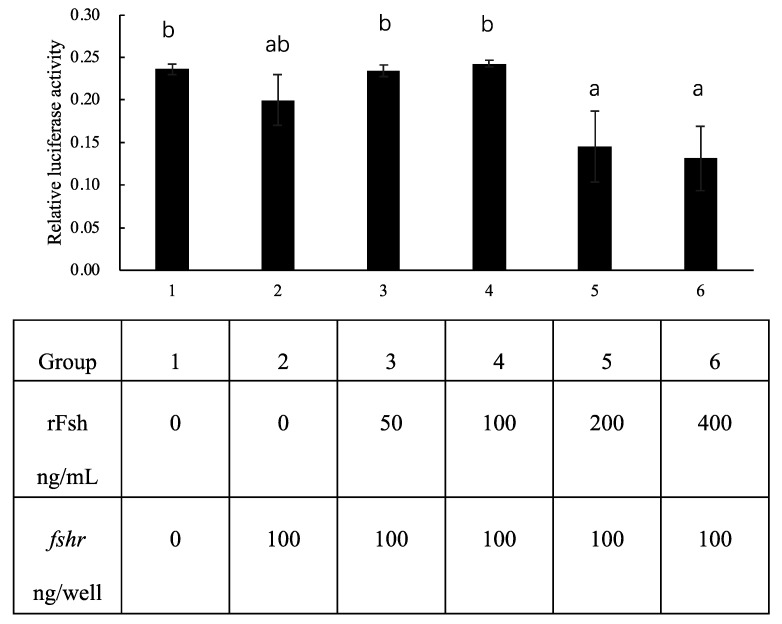
Effects of different concentrations of rFsh on Nile tilapia *cyp19a1a* promoter activity in HEK293T cells. *fshr* (0–100 ng) expression vectors were transfected into HEK293T cells with the Nile tilapia *cyp19a1a* promoter construct (500 ng/well), and the pRL-TK control vector (100 ng/well). Firefly and *Renilla* luciferase activities were measured 48 h after transfection. Relative luciferase activity was calculated by dividing the firefly luciferase activity by the *Renilla* luciferase activity. Data are presented as the mean ± standard error (SEM) (*n* ≥ 3). Significant differences at *p* < 0.05 are indicated by different letters.

**Figure 7 ijms-26-05376-f007:**
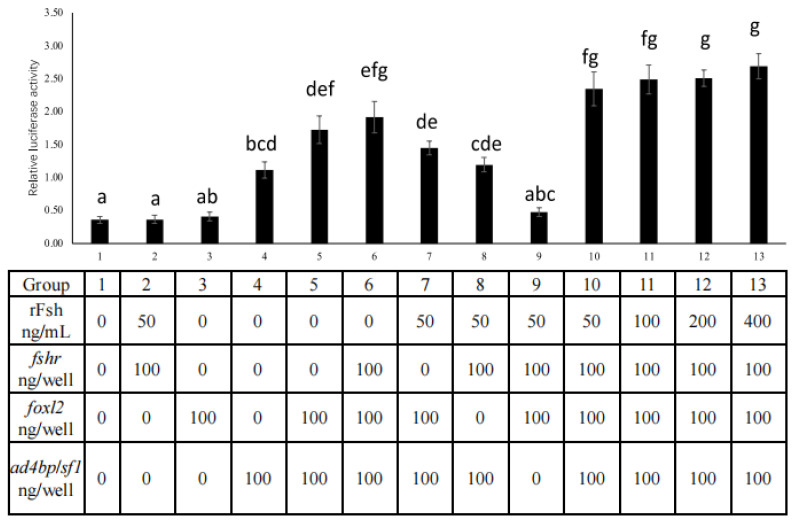
Effects of FSH signaling, *foxl2* and *ad4bp*/*sf1* on Nile tilapia *cyp19a1a* promoter activity in HEK293T cells. FSH signaling, alone or together with *foxl2* and *ad4bp*/*sf1* expression vectors, were co-transfected into HEK293T cells with the Nile tilapia *cyp19a1a* promoter construct (500 ng/well) and the pRL-TK control vector (100 ng/well). Firefly and *Renilla* luciferase activities were measured 48 h after transfection. Relative luciferase activity was calculated by dividing the firefly luciferase activity by the *Renilla* luciferase activity. Data are presented as the mean ± standard error (SEM) (*n* ≥ 3). Significant differences at *p* < 0.05 are indicated by different letters.

**Table 1 ijms-26-05376-t001:** Sequences of the qPCR and promoter assay primers and morpholinos used in this study.

Experiment	Name	Sequence
qPCR	*fshr*-sense	5′-CGGGCTGAGGATTTTTCCA-3′
	*fshr*-antisense	5′-TGTTGTCCTGAAGATCCAGCAG-3′
	*lhr*-sense	5′-CAGTGCAGAATATCAACAGCCTGA-3′
	*lhr*-antisense	5′-TGTTAGAGATGCTCAAATATTCCAGCTT-3′
*cyp19a1a promoter assay*	*cyp19a1a* promoter fragment-sense	5′-TACGCGTGCTAGCCCCACAGCCTCCATTCACCA-3′
	*cyp19a1a* promoter fragment-antisense	5′-GCAGATCTCGAGCCCGAGAAGGGTGATGATGTAGAAC-3′
	*foxl2*-ORF-sense	5′-TAGGCTAGCCTCGAGATGATGGCCACTTACCAAAAC-3′
	*foxl2*-ORF-antisense	5′-TACCACGCGTGAATTTCAAATATCAATCCTCGTGTGTAAC-3′
	*ad4bp/sf1*-ORF-sense	5′-TAGGCTAGCCTCGAGATGTTGGGAGACAAGGCTCA-3′
	*ad4bp/sf1*-ORF-antisense	5′-TACCACGCGTGAATTTCACACACACGCCCTCTTAG-3′
MO-knock down	*fshr*-MO	5′-GTGTCATTACCAGCATCATTTCAGT-3′
	*fshr*-MO control	5′-GTcTgATTACgAGCATCATTTgAcT-3′
	*fsh*-MO	5′-ATCCTCTGCCGGATGCACTACACGC-3′

Note: the underlined bases are complementary to ATG start codon.

## Data Availability

The original contributions presented in this study are included in the article. Further inquiries can be directed to the corresponding author.

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
