# Peer review of "The Potential Role of Gonadotropic Hormones and Their Receptors in Sex Differentiation of Nile Tilapia, Oreochromis niloticus"

_ijms, 2025, doi:10.3390/ijms26115376_

Round 1
Reviewer 1 Report
Comments and Suggestions for Authors
This study presents a valuable investigation into the role of FSH signaling in ovarian differentiation in Nile tilapia. The use of both knockdown approaches and transcriptional assays strengthens the conclusions. Your finding that FSH signaling may not be critical but potentially plays a compensatory role in regulating cyp19a1a expression is particularly interesting and adds nuance to our understanding of teleost sex differentiation.
General Comments
-
The author mentions Nile tilapia as an ideal model for teleosts. Please consider elaborating on why it is considered ideal, or rephrase the sentence for clarity.
-
What is the rationale or broader significance of this study? Are there any potential applications in selective breeding or aquaculture?
-
Please include scale bars and corresponding values in all figure panels.
-
Figures 4 and 5 are unclear and appear blurry. Please provide higher-resolution images or improve figure quality.
Abstract
The abstract is generally well written and clearly outlines the objectives and findings of the study. However, the opening sentence could be revised for a stronger and more engaging start.
Introduction
-
Consider adding a few sentences on the reproduction and breeding biology of Nile tilapia to provide context.
-
Please include more references discussing the comparative responses of vertebrates to gonadotropins (GTHs) to better position the study within existing literature.
Materials and Methods
-
The dosing strategy appears appropriate; however, inclusion of dose–response data would strengthen the conclusions regarding treatment efficacy.
-
Please include information on primer efficiency to validate the qPCR results.
-
The knockdown experiments should be described in more detail, especially regarding their design, controls, and verification.
Results
-
The results are well documented. However, Figures 4 and 5 need improvement in visibility and clarity. Please provide higher-quality versions to support the interpretation of results.
Discussion and Conclusions
-
The discussion and conclusions are clearly written and are acceptable in their current form.
Author Response
Responses to the reviewer #1:
>The author mentions Nile tilapia as an ideal model for teleosts. Please consider elaborating on why it is considered ideal, or rephrase the sentence for clarity.
We thank you for your evaluation of our paper. We appreciate your efforts to improve the quality of our manuscript by providing valuable suggestions and comments. Thanks for your suggestion, we added some sentences to explain the reason why Nile tilapia is an ideal for teleosts. (Line 39-41)
>What is the rationale or broader significance of this study? Are there any potential applications in selective breeding or aquaculture?
Thanks for your suggestion. We explain the importance of sex control in aquaculture. And explain the application of this study in aquaculture. (Line 31-34, and Line 94-98)
>Please include scale bars and corresponding values in all figure panels.
Thanks for your suggestion. Scale bars were included in all figures.
>Figures 4 and 5 are unclear and appear blurry. Please provide higher-resolution images or improve figure quality.
Thanks for your suggestion. High-quality Figures 4 and 5 were provided.
Abstract
>The abstract is generally well written and clearly outlines the objectives and findings of the study. However, the opening sentence could be revised for a stronger and more engaging start.
Thanks for your suggestion. We wrote the opening sentence in the abstract again. (Line 13 and 14)
Introduction
>Consider adding a few sentences on the reproduction and breeding biology of Nile tilapia to provide context.
Thanks for your suggestion. More information about the reproduction and breeding biology of Nile tilapia was added in the introduction. (Line 42-45)
>Please include more references discussing the comparative responses of vertebrates to gonadotropins (GTHs) to better position the study within existing literature.
Thanks for your suggestion. The more detailed information about the types of cells expressing fshr in mammals and fish was compared to discuss the comparative responses of vertebrates to FSH signaling. And the function of LH signaling in fish was explained. (Line 78-80, and Line 92-93)
Materials and Methods
>The dosing strategy appears appropriate; however, inclusion of dose–response data would strengthen the conclusions regarding treatment efficacy.
The dosing strategy was designed based on the results of a previous study. Gao, H.; Arai, T.; Aranyakanont, C.; Li, D.; Tada, M.; Ijiri, S. Role of Follicle-Stimulating Hormone in Gonadal Sex Differentiation via Expression of Steroidogenic Enzymes in Nile tilapia, Oreochromis niloticus. Fish. Sci. 2025, 91, 13–23.
In Figure 7, the transcriptional activity induced by foxl2 and ad4bp/Sf1 (group 5) was significantly elevated with rFsh signaling stimulation (groups 12 and 13); however, the level of elevation by FSH stimulation was not very strong. Therefore, we suggest that FSH signaling is involved in cyp19a1a transcription with foxl2 and ad4bp/Sf1.
>Please include information on primer efficiency to validate the qPCR results.
Thanks for your suggestion. In this study, the same primers were used as in the previous study. Yan, H.; Ijiri, S.; Wu, Q.; Kobayashi, T.; Li, S.; Nakaseko, T.; Adachi, S.; Nagahama, Y. Expression Patterns of Gonadotropin Hormones and Their Receptors During Early Sexual Differentiation in Nile tilapia Oreochromis niloticus. Biol. Reprod. 2012, 87.
>The knockdown experiments should be described in more detail, especially regarding their design, controls, and verification.
Thanks for your suggestion. The methods, especially regarding design, controls, and verification in the knock-down experiment, were rewritten to make it easier to understand. (Line 432-437)
Results
>The results are well documented. However, Figures 4 and 5 need improvement in visibility and clarity. Please provide higher-quality versions to support the interpretation of results.
Thanks for your suggestion. High-quality Figures 4 and 5 were provided.
Discussion and Conclusions
>The discussion and conclusions are clearly written and are acceptable in their current form.
We appreciate your thoughtful suggestions and insights.
Reviewer 2 Report
Comments and Suggestions for Authors
Authors present a well done study on the endocrine and molecular mechanisms of sex differentiation in Nile tilapia fish. I miss a list of abbreviations used in the manuscript (genes, enzymes, hormones and other chemicals, buffers etc.) for non-specialists. I also miss a conclusion summarizing the most important results of the present study compared to those from former experiments.
Other corrections:
- Legends to Figs 1and 2: authors say twice that bars present SEM; delete it once
- Legend to Fig. 6: Effects of different concentrations...
- Line 199 and others: Renilla is a genus name and should be italicsed in the entire manuscript
- Fig. 7: use either l or L for liter in the whole manuscript, not mixed
- lines 388 and 389: L cannot be correct here, Dto. in lines 397 .399
- lines 426: umoles as an amount of injection?
- line 464: there is no Table 2 in the manuscript
- line 515 and others in the References: names of animals must be given correctly; the second part in lowercase letters
- line 578: Proc. Natl. Acad. Sci. USA
Author Response
Responses to the reviewer #2:
>Authors present a well done study on the endocrine and molecular mechanisms of sex differentiation in Nile tilapia fish. I miss a list of abbreviations used in the manuscript (genes, enzymes, hormones and other chemicals, buffers etc.) for non-specialists. I also miss a conclusion summarizing the most important results of the present study compared to those from former experiments.
Thanks for your suggestion. We added the list of abbreviations. The most important results in this study were summarized and compared with the previous study. (Line 363-370)
Other corrections:
Legends to Figs 1and 2: authors say twice that bars present SEM; delete it once
Thanks for your suggestion. I deleted SEM once. (Figure 1 and 2)
Legend to Fig. 6: Effects of different concentrations...
Thanks for your suggestion. The word different was written. (Figure 6)
Line 199 and others: Renilla is a genus name and should be italicsed in the entire manuscript
Thanks for your suggestion. All Renilla was italicised. (Line 214, 215, 241, 243, 505, 507, and 508)
Fig. 7: use either l or L for liter in the whole manuscript, not mixed
Thanks for your suggestion. l has been changed into L. (Figure 7)
lines 388 and 389: L cannot be correct here, Dto. in lines 397 .399
Thanks for your suggestion. L has been changed into µL. (Line 406, 407, 416, and 417)
lines 426: umoles as an amount of injection?
Thanks for your question. 0.87 umoles/embryo is the amount of injection. (Line 446)
line 464: there is no Table 2 in the manuscript
Thanks for your suggestion. Table 2 has been changed into Table 1. (Line 484)
line 515 and others in the References: names of animals must be given correctly; the second part in lowercase letters
Thanks for your suggestion. Names of animals were written correctly. (Line 530, 539, 544, 559, 571, 573, 576, 595, 597, 600, 614, and 616)
line 578: Proc. Natl. Acad. Sci. USA
Thanks for your suggestion. It was written correctly. (Line 620)